# The Role of the Universally Conserved ATPase YchF/Ola1 in Translation Regulation during Cellular Stress

**DOI:** 10.3390/microorganisms10010014

**Published:** 2021-12-23

**Authors:** Victoria Landwehr, Martin Milanov, Jiang Hong, Hans-Georg Koch

**Affiliations:** 1Institut für Biochemie und Molekularbiologie, ZBMZ, Fakultät für Medizin, Albert-Ludwigs-Universität Freiburg, 79104 Freiburg, Germany; victoria.Landwehr@t-online.de (V.L.); Martin.Milanov@biochemie.uni-freiburg.de (M.M.); hongj.jackie@gmail.com (J.H.); 2Fakultät für Biologie, Albert-Ludwigs-Universität Freiburg, 79104 Freiburg, Germany; 3Spemann-Graduate School of Biology and Medicine, Albert-Ludwigs-Universität Freiburg, 79104 Freiburg, Germany

**Keywords:** YchF, Ola1, ribosomes, stress response, GTPases, ATPases, translation initiation, leaderless mRNA

## Abstract

The ability to respond to metabolic or environmental changes is an essential feature in all cells and involves both transcriptional and translational regulators that adjust the metabolic activity to fluctuating conditions. While transcriptional regulation has been studied in detail, the important role of the ribosome as an additional player in regulating gene expression is only beginning to emerge. Ribosome-interacting proteins are central to this translational regulation and include universally conserved ribosome interacting proteins, such as the ATPase YchF (Ola1 in eukaryotes). In both eukaryotes and bacteria, the cellular concentrations of YchF/Ola1 determine the ability to cope with different stress conditions and are linked to several pathologies in humans. The available data indicate that YchF/Ola1 regulates the stress response via controlling non-canonical translation initiation and via protein degradation. Although the molecular mechanisms appear to be different between bacteria and eukaryotes, increased non-canonical translation initiation is a common consequence of YchF/Ola1 regulated translational control in *E. coli* and *H. sapiens*. In this review, we summarize recent insights into the role of the universally conserved ATPase YchF/Ola1 in adapting translation to unfavourable conditions.

## 1. Introduction

All cells, in particular unicellular organisms, are permanently challenged by rapid environmental changes and therefore depend on sophisticated mechanisms for adjusting their metabolic activity to external cues. In bacteria, a major hallmark of these adjustments are transcriptional regulators, such as two-component systems [1,2] or various sigma factors [3] that rapidly adapt gene expression in response to, e.g., temperature shifts, pH shifts or nutrient limitation. In addition to these well-studied transcriptional responses, bacteria also adjust their metabolism by modifying the translational activity of the ribosome [4,5,6]. Examples are the ribosome modulation factor (RMF) and the hibernation-promoting factor (HPF), which are produced during the stationary phase in *Escherichia coli* [7,8,9]. Both proteins sequentially interact with 70S ribosomes and convert them into translationally inactive dimeric 100S ribosomes. This rapidly reversible inactivation of ribosomes is considered to prevent ribosomal damage while simultaneously reducing the production of damage-prone translation products. Another example for translational regulation is the so-called stringent response, which is initiated during nutrient limitation, e.g., shortage of amino acids [10,11]. The presence of an uncharged tRNA in the A-site of the ribosome serves as a signal for recruiting the GTP-pyrophosphokinase RelA to the ribosome. Subsequently, RelA phosphorylates GTP to the hyper-phosphorylated guanine-nucleotides pppGpp or ppGpp, collectively called alarmones [12,13]. The gradual increase in these alarmones allosterically regulates the specificity of RNA-polymerase, leading to reduced synthesis of ribosomal RNA and ribosomal proteins. In addition, alarmones competitively inhibit GTPases, such as initiation factor 2 (IF2), which further attenuates translation during substrate limitation.

Ribosome inactivation by dimer formation as well as (p)ppGpp production are typical stress responses in bacteria and not found in eukaryotes, with the exception of bacteria-derived organelles, such as chloroplasts [14,15]. However, there are also universally conserved stress response proteins acting on the ribosome, such as the P-loop ATPase YchF (Ola1 in eukaryotes) [16,17]. Although YchF belongs to the translation-factor-related (TRAFAC) class of GTPases, it preferentially hydrolyses ATP rather than GTP [18,19]. Deletion or depletion of *ychF*/*ola1* in *E. coli*, *Oryza sativa* or *Homo sapiens* is associated with increased stress resistance, suggesting a role of YchF/Ola1 as a negative regulator of stress response pathways [16,20,21,22]. YchF is furthermore considered as a constituent of the minimal translation machinery in bacteria [23]. In this review, we summarize recent developments that shed light on the molecular functions of YchF/Ola1 in regulating the response to unfavourable cellular and environmental conditions.

## 2. Structure of YchF/Ola1 and Its Interaction with Ribosomes

YchF/Ola1 is a highly conserved three-domain protein (Figure 1), sharing 45% sequence identity and 62% sequence similarity between human Ola1 and *E. coli* YchF (Figure 2). YchF/Ola1 contains four conserved G-motifs (G1–G4), which defines YchF/Ola1 as a member of the large group of P-loop NTPases [16,17,24]. The G1–G3 motifs are located in the N-terminal G-domain, which also contains a conserved serine residue (S16 in *E. coli*) and a widely conserved cysteine residue (C35 in *E. coli*), important for phosphorylation [25] and dimerization [26], respectively (Figure 2).

The G4 motif of YchF/Ola1 in most species differs from the canonical G4 motif in P-loop GTPases and is composed of NxxE instead of NKxD (Figure 2), which is likely responsible for the altered nucleotide specificity of YchF/Ola1 [19]. An exception is YchF of *Haloferax volcanii*, which contains the typical NKxD motif (Figure 2), but the nucleotide specificity of *H. volcanii* YchF has not been tested so far. The G-domain is followed by a helical domain (coiled-coil domain), which forms together with the C-terminal TGS (ThrRS, GTPase, SpoT)-domain, a large, approx. 20 Å wide positively charged cleft, potentially favouring RNA interactions (Figure 1) [19,27]. The exact roles of the helical and the TGS-domains are unknown. In most proteobacteria, the helical domain of YchF contains two conserved additional cysteine residues (C140, C168 in *E. coli*), which are perfectly positioned for an intramolecular disulphide-bridge [26] (Figure 1 and Figure 2). In *E. coli*, the helical domain was also shown to interact with thioredoxin 1 [26] and thus could act as a redox switch. The TGS-domain is present in many RNA-binding proteins, such as ObgE, Threonyl-tRNA synthetase, (p)ppGpp synthase/hydrolase SpoT and the GTP-pyrophosphokinase RelA [12,28,29]. TGS-domains belong to the beta-grasp fold superfamily, which also include ubiquitin-like proteins [30], associated with post-translational modifications and proteolysis [31,32].

Within the TRAFAC class of GTPases, YchF/Ola1 belongs to the Obg-like protein family, which is associated with ribosome biogenesis and translational control [16,17,33]. In *E. coli* and other enterobacteria, *ychF* is encoded in one operon together with the gene for the essential peptidyl-tRNA hydrolase Pth [34], which catalyses the removal of peptides from peptidyl-tRNAs that have been prematurely released from the ribosome [35]. This also points to a ribosome-associated role of YchF. Thus, several studies have analysed ribosome binding of YchF/Ola1 and its potential effect on ribosome biogenesis in different organisms [36,37,38,39,40,41,42] (Table 1). A putative role in ribosome biogenesis has only been suggested for *Arabidopsis thaliana* AtYchF, based on the homology to known ribosome assembly factors, such as ObgE [40], but there is no experimental evidence for an involvement of YchF/Ola1 in ribosome biogenesis so far. Studies in *E. coli* using a Δ*ychF* strain or an *ychF*-overexpressing strain did not show any significant effect on the presence and quantities of the 30S, 50S, 70S or polysome fractions [43,44].

On the other hand, ribosome binding of YchF/Ola1 has been experimentally determined in *E. coli* [20,36,37,42], yeast [41,45], *Trypanosoma* [38], plants [40,46] and humans [39] (Table 1). Still, the specific binding site of YchF/Ola1 on the ribosome is unknown. Co-fractionation studies indicate co-migration of YchF/Ola1 with the 30(40)S subunit, the 50(60)S subunit, 70(80)S ribosomes and polysomes [37,38,39,42]. Co-immune precipitation studies in *Trypanosoma* further indicate that YchF binds to the 60S ribosomal protein uL24 at the ribosomal tunnel exit and to the 40S ribosomal protein eS7, which is implicated in rRNA processing and translational fidelity [38,47]. In vivo site-directed cross-linking studies confirmed contacts of YchF with both the 30S and the 50S ribosomal subunits in *E. coli* (Figure 3) [42]. The large majority of these contacts were found on the 30S subunit, e.g., uS5, uS8, uS11 and bS21, which are located at the interface between the 30S and 50S subunits, close to the mRNA entrance site (Figure 3) [42]. A similar contact of Ola1 to the 40S–60S interface is deduced from the observation that the over-production of Ola1 rescues the phenotypes associated with deletions of the ribosomal proteins uS17 and eS7 in yeast [45]. Both proteins are located at the mRNA corridor between the 40S and 60S subunits and are required for translational fidelity [48]. The cross-linking approach in *E. coli* also detected uL29 as a potential contact site for YchF [42]. Similar to uL24, uL29 is located at the ribosomal tunnel exit of the 50S subunit [49], suggesting that YchF/Ola1 can interact with both subunits of the ribosome. The N-terminal ATPase domain of YchF appears to serve as a major contact site to the ribosome [42], which supports the observation that ribosome binding stimulates ATPase activity [36].

The available data so far do not support the presence of a single binding site for YchF on the ribosome. In *E. coli*, YchF is largely sub-stoichiometric to ribosomes [36,50] and only approx. 10% of all ribosomes can be in contact with YchF in vivo. Thus, most studies on ribosome binding were performed under YchF over-producing conditions, which likely influenced the YchF-ribosome interaction. This is supported by the observation that in *E. coli* wild type cells, YchF was exclusively found in the 30S ribosomal fraction, while in YchF-over-producing cells it was found in the 30S, 50S, 70S and polysome fractions [42]. In accordance with the available functional data (see below) the 30S/40S ribosomal subunit seems to be the preferred binding site for YchF/Ola1, while binding to the 50S/60S subunit is probably only transient [36,38,39,42].

**Table 1 microorganisms-10-00014-t001:** Function of YchF/Ola1 in different species.

Organism	Ribosome Binding	Translation	Proteolysis	Cellular Adaptation ^1^	References
*Escherichia coli*	+	+	-	+	[20,36,42]
*Brucella melitensis*	n.a. ^2^	n.a.	n.a.	+	[51,52]
*Haemophilus influenzeae*	n.a.	n.a.	n.a.	+	[27]
*Streptococcus pneumonia*	n.a.	n.a.	n.a.	+	[53]
*Vibrio vulnificus*	n.a.	n.a.	n.a.	+	[54]
*Saccharomyces cerevisiae*	+	+	+	+	[41,45,55]
*Arabidopsis thaliana*	+	n.a.	n.a.	+	[22,40]
*Oriza sativa*	+	n.a.	n.a.	+	[22,46,56]
*Trypanosoma cruzi*	+	n.a.	+	n.a.	[38]
*Mus musculus*	+	+	+	+	[39,57,58]
*Homo sapiens*	+	+	+	+	[39,59,60]

^1^ cellular adaptation includes stress response, metabolic alterations, virulence and other adaptation pathways that are influenced by YchF/Ola1; see text for details. ^2^ n.a. = not analysed/ no data available.

## 3. Cellular Roles of YchF/Ola1 in Stress Response

Multiple studies have linked YchF/Ola1 to stress response pathways [20,21,22,26,39]. Intriguingly, the absence/depletion of YchF/Ola1 leads to a gain of function phenotype in some cases, i.e., the corresponding strains tolerate non-favourable conditions much better than wild type cells. This is shown for the Δ*ychF* strain of *E. coli*, which is resistant against the replication inhibitor hydroxyurea and the translational inhibitor fusidic acid [42]. The reversal of these phenotypes is observed by providing just *ychF* in trans, demonstrating that the observed phenotypes are not caused by a reduced expression of the *pth* gene, which forms a bicistronic transcript together with *ychF* [20,26,42]. Other examples for a stress-related function of YchF/Ola1 is the increased resistance of Ola1-depleted cells against DNA-damaging agents [61,62] or the increased salt sensitivity of plants, when YchF is over-produced [22,46,56]. Even though in some cases the absence of YchF/Ola1 is beneficial for cells encountering stress conditions, the Δ*ychF* strain is rapidly outcompeted by wild type *E. coli* cells under non-stress conditions [42], pointing to an important role of YchF in cellular metabolism. This is also supported by the cold sensitivity of the Δ*ychF* strain. The cold-sensitive phenotype is rescued by a plasmid-borne copy of wild type *ychF* but not by the ATPase-deficient mutant [20], suggesting that ATP hydrolysis is required for growth at sub-optimal temperatures. Cold sensitivity in bacteria is frequently associated with changes in membrane composition, e.g., alterations in lipid or protein composition [63,64]. Whether the absence of YchF has any impact on lipid or protein composition of the bacterial membrane is currently unknown, but the reduced iron uptake in Δ*ychF* strains of *Vibrio vulnificus* and *Brucella melitensis* could support possible membrane alterations [51,54].

A common phenotype of *ychF*-deletion strains of *E. coli* and Ola1-depleted human cells is their increased resistance to reactive oxygen species (ROS), such as peroxides, and to thiol-depleting agents, such as diamide [20,21,26]. In *E. coli*, *ychF* expression is downregulated when cells are exposed to H_2_O_2_ [20], indicating that lowering the YchF concentration is a physiological response against ROS accumulation. Accumulating H_2_O_2_ activates the oxidative stress-responsive transcription factor OxyR [65], which recognizes a specific binding motif within the 5′-UTR of *ychF* and prevents transcription initiation [20]. The transcriptional downregulation of *ychF* in *E. coli* is likely important during long-term ROS exposure [42], e.g., when cells transition from early exponential phase to late exponential and stationary phase, which is accompanied by increased ROS production [66]. *E. coli* YchF is also a substrate of the AAA-protease Lon [67], which is required for cellular homeostasis and for stress-induced developmental changes [68]. Thus, Lon-dependent proteolysis of YchF during stress conditions could additionally contribute to the decline of the YchF levels.

*E. coli* YchF co-purifies with catalases and many antioxidant proteins, including thioredoxin or glutaredoxin, suggesting that YchF could act as a direct inhibitor of antioxidant enzymes [20,26]. This is supported by a reduction in catalase activity when YchF is over-produced [20]. However, YchF does not inhibit the catalase activity of KatG in vitro [42], indicating that the reduced catalase activity is likely an indirect effect. On the other hand, the interaction of YchF with thioredoxin was shown to be required for dissociating the YchF dimer, which is formed by ROS-induced oxidation of the conserved Cys35-residue and intermolecular disulphide-bridge formation [26]. Within the YchF dimer, the ATP binding site is covered and hence dimerization prevents ATP hydrolysis [26]. The increased ATPase activity of YchF in the presence of ribosomes [36] thus indicates that monomeric YchF interacts with the ribosome. Cys35 is conserved in many species (Figure 2), suggesting that inactivation via ROS-induced disulphide-bridge formation is not limited to *E. coli* [26,41]. Overproduction of YchF increases ROS sensitivity, but this is not observed when the ATP-binding site in YchF is mutated, showing that ATP hydrolysis by the YchF monomer is important for regulating the oxidative stress response [20].

In bacteria and humans, reduced YchF/Ola1 levels increase the resistance against ROS and DNA-damaging agents, suggesting a role of YchF/Ola1 as negative regulator of these stress-response pathways. In contrast, Ola1 acts as positive regulator of the heat-shock response in humans and yeast [41,69]. Upon depletion of Ola1 in human cell lines, the levels of the heat-shock protein 70 (Hsp70) decrease, leading to increased temperature-induced cell death [69]. Ola1 was shown to bind to the C-terminus of Hsp70, thereby preventing its ubiquitination and proteasomal degradation [69]. The enhanced ubiquitination and degradation of Hsp70 in the absence of Ola1, but not of other heat-shock proteins [69], is likely responsible for the temperature sensitivity of Ola1-depleted human cells. A potential link between temperature sensitivity and peroxide resistance is provided by the increased degradation of the H_2_O_2_-generating enzyme superoxide dismutase when Hsp70 levels are reduced in the absence of Ola1 [59]. Due to the reduced intracellular H_2_O_2_ production, cells are potentially more resistant towards externally added H_2_O_2_.

Enhanced ubiquitination is also observed in Δ*ola1* cells of yeast. Here, the absence of Ola1 increased the ubiquitination of heat-shock induced protein aggregates [41], and caused an up-regulation of heat-protective proteins, such as Hsp12, Hsp42 and the Vtc complex [41], which is involved in the synthesis of the chemical chaperone polyphosphate [70,71]. Yeast Ola1 was shown to assemble into heat-induced stress granules [72] and to be released from these granules upon stress relief [41]. In the absence of Ola1, recovery from heat stress is delayed because translation initiation is reduced [41] and it was suggested that Ola1 could serve as an early heat-shock responder that regulates proteostasis for preventing damage or misfolding of proteins [41]. A putative role of Ola1 in regulating ubiquitin-dependent protein degradation is supported by the observed interaction of Ola1/YchF with subunits of the proteasome [38,41,55]. The C-terminal TGS-domain of Ola1/YchF is a potential candidate for the interaction with the proteasome, due to its structural similarity to ubiquitin-like proteins [30].

The importance of well-balanced Ola1-levels is further highlighted by the observation that Ola1 is up-regulated in various human tumours, such as colorectal carcinoma and ovarian and lung cancer [61]. Clinical progression and prognosis of these tumours is influenced by the cellular Ola1 concentrations [57]. Tumour formation is at least partially linked to changes in Ser/Thr phosphorylation of some cancer-related proteins upon increased Ola1-levels. This includes glycogen synthase kinase-3 (GSK-3) [73], focal adhesion kinases (FAKs) [74], peptidyl-prolyl isomerase Pin1 [73,75] and eukaryotic initiation factor 2 (eIF2) [39]. Involvement of Ola1 in breast and ovarian cancer is likely connected to its interaction with a protein complex consisting of BRCA1 (breast cancer associated gene 1), BARD1 (BRCA1-associated Ring domain protein) and γ-tubulin [60,76]. This complex is associated with the centrosome and important for chromosome stability [77,78,79]. The BRCA1-BARD1 complex acts as a Ring-type E3 ubiquitin ligase, catalysing the ubiquitination of multiple substrate proteins [80]. The ubiquitin-like C-terminal TGS-domain of Ola1 is required for the interaction with the N-terminal domains of BRCA1 and BARD1 [60,78], which supports the functional importance of the TGS-domain in Ola1/YchF.

## 4. The Role of YchF/Ola1 in Translation

The available data support the primary role of YchF/Ola1 in regulating the proteostasis network when cells encounter stress conditions [81,82]. This includes translational regulation at the level of translation initiation [39,41,42,58] and regulation of protein degradation [59,69,73]. Observations such as decreased cell migration, increased cell matrix adhesion [57,62] and reduced cell cycle progression [58] in the absence of Ola1 are likely either directly or indirectly linked to its role in proteostasis regulation.

Translation is primarily controlled at the energy-consuming initiation step, which is significantly different between eukaryotes and bacteria [83,84,85]. In eukaryotes, initiation starts with the formation of a ternary complex, consisting of eIF2-GTP, and the aminoacylated tRNAi^Met^ (eIF2-GTP-Met-tRNAi^Met^; Figure 4A). In conjunction with the 40S ribosomal subunit and additional initiation factors (eIF1, eIF2A, eIF3 and eIF5) the ternary complex forms a 43S pre-initiation complex, which binds the activated mRNA. mRNA activation involves specific recognition of the m7G-Cap structure at the 5’end of the mRNA by the eIF4F complex. The resulting 48S complex then scans the mRNA for identifying the start codon. Once the Met-tRNAi^Met^ has recognized the AUG start codon, eIF2 and other initiation factors are released, enabling 60S subunit binding, resulting in the elongation competent 80S ribosome [86,87]. This Cap-dependent initiation is primarily regulated at the level of ternary complex formation via inactivation of eIF2 [88].

Ola1 was shown to prevent ternary complex formation via binding to eIF2 [39], which would subsequently reduce protein synthesis (Figure 4A). Reduced formation of the ternary complex is a well-studied consequence of the activated integrated stress response (ISR) [89]. Activation of multiple kinases by the ISR leads to stress-induced inhibition of eIF2 via phosphorylation [86]. Ola1 is suggested to provide an alternative mechanism for reducing ternary complex formation by stabilizing the eIF2-GDP complex (Figure 4A), which would prevent further initiation events [39,86]. Although Cap-dependent translation is common under non-stress conditions, eukaryotic cells also engage Cap-independent translation mechanisms when they are exposed to stress [86,90,91]. One example for a Cap-independent translation mechanism is the translation of the cAMP-dependent transcription factor ATF4, which is produced by a translation re-initiation mechanism in response to eIF2 phosphorylation [90,92,93]. Stimulated ATF4 production as a consequence of reduced ternary complex formation in the presence of Ola1 would then promote transcription of genes linked to oxidative stress, ER stress or nutrient limitation [94,95,96].

Thus, the available data suggest that upon stress conditions, increased Ola1-levels attenuate canonical translation initiation, while simultaneously favouring alternative translation initiation mechanisms, which stimulates cell survival under stress conditions (Figure 4A). This assumption is further supported by the observation that Ola1 did not inhibit the Cap-independent translation of viral proteins [39]. Translation of viral proteins is often initiated by mRNA elements called internal ribosome binding sites (IRES) [97], but IRES-based initiation is also frequently used for translating stress response proteins [98].

Translation initiation in bacteria deviates from initiation in eukaryotes and starts with the formation of an instable 30S pre-initiation complex. This complex contains the initiation factors (IF) IF1, IF2 and IF3, the fMet-tRNA^fMet^ and the mRNA, which is bound via its Shine-Dalgarno sequence to the anti-Shine-Dalgarno sequence within the 16S rRNA of the 30S subunit [85,99]. The recognition of the AUG start codon by the fMet-tRNA^fMet^ induces a conformational change that converts the 30S pre-initiation complex into the 30S initiation complex. Due to the conformational change, IF3 is destabilized and dissociates, allowing the recruitment of the 50S subunit and transition of the 30S initiation complex to the 70S initiation complex [100]. Dissociation of IF3 also weakens ribosome binding of IF1 and IF2, which dissociate during transition to the 70S initiation complex [85].

Similar to eukaryotes, bacteria engage alternative translation initiation mechanisms, and they are particularly important for those mRNAs that lack the Shine-Dalgarno sequence. The occurrence of these leaderless mRNAs (lmRNAs) is highly variable between species [101,102]. They are highly abundant in *Deinococcus deserti* (47% of all mRNAs) and their translation is linked to the extreme stress resistance of this species [103]. lmRNAs are also frequently observed in *Mycobacterium spec.* (33% in *M. avium*) and expressed when cells encounter stress conditions [104,105,106]. In the halophilic archaeon *Haloferax volcanii* about 72% of all mRNAs are leaderless [107,108,109]. In proteobacteria, such as *E. coli*, the percentage of naturally occurring lmRNAs is rather low (0.7% in *E. coli* BW25113) [101,110], but they can be generated by post-transcriptional processing of canonical mRNAs, e.g., by the ribonuclease MazF [111,112]. MazF constitutes the toxin of the MazEF toxin-antitoxin system [113,114]. Upon stress conditions, the labile anti-toxin MazE is proteolized and MazF is released for mRNA degradation [5,113,115]. Although MazF predominantly degrades mRNAs completely [116,117], it can also generate lmRNAs by cleaving off the Shine-Dalgarno sequence [111]. Among the predicted MazF targets are many mRNAs encoding for stress response proteins [112], which supports the idea that selective lmRNA translation is an important cornerstone of stress adaptation [5,118].

Due to the absence of the Shine-Dalgarno sequence, the AUG initiation codon at the very 5′-end of the mRNA serves as a major signal for ribosome binding to these lmRNAs. However, in some bacterial species, including Bacteroidetes and Cyanobacteria, mRNAs contain a 5′-leader sequence without clearly recognizable Shine-Dalgarno motifs and their initiation is at least partially determined by the RNA structure around the initiation codon [102,119,120,121]. Most studies indicate that translation of mRNAs starting with the AUG start codon is initiated on 70S ribosomes in bacteria [101]. This is deduced from the observation that 70S ribosomes bind lmRNAs with higher affinity than 30S subunits [122] and that the stability of the 70S initiation complexes increased in the presence of lmRNA [123]. Translation of canonical mRNAs, but not that of lmRNAs, is inhibited by chemical cross-linking or by inactivation of ribosome recycling factors, which prevent dissociation of the 70S ribosome [124]. This further supports the hypothesis that translation of lmRNAs is initiated on 70S ribosomes.

Absence of YchF promotes translation of lmRNAs and reduces the translation of canonical mRNAs [42]. This is in line with reduced global protein synthesis of the Δ*ychF* strain and the observation that Δ*ychF* cells are outcompeted by wild type cells under non-stress conditions [42]. It was recently shown that IF3 transiently occupies a second binding site on the 50S ribosomal subunit, which might be important for the translation initiation of lmRNAs [125]. IF3 and YchF occupy overlapping binding sites on the 30S subunit [42,126] and both also interact with the 50S subunit [42,125]. A direct contact between YchF and IF3 was furthermore demonstrated by cross-linking in vivo [42]. Thus, one attractive model is that YchF controls lmRNA translation by regulating IF3 binding to either the 30S or 50S ribosomal subunit (Figure 4B). Under normal growth conditions, YchF supports IF3 binding to the 30S subunit and thus favours canonical translation. When cells enter the stationary phase or encounter stress conditions, the YchF levels drop [42], while IF3 levels increase [127]. This enhances the interaction of IF3 with the 50S subunit and promotes lmRNA translation. This model is in line with recent data showing enhanced lmRNA translation under oxidative stress conditions [128], which are associated with reduced YchF levels [20].

In essence, a decrease in YchF and an increase in IF3 during the stationary phase or stress conditions would prepare the translation machinery for the increased lmRNAs accumulation that occurs due to increased MazF-dependent cleavage of the Shine-Dalgarno sequence under these conditions [111,112] (Figure 4B). Increased lmRNA translation in the absence of YchF possibly also explains the cold-sensitive phenotype of the *E. coli* Δ*ychF* strain [20]. During cold shock, IF2 levels increase and IF2 accumulation stimulates lmRNA translation at the expense of canonical translation [63,127]. Further stimulation of lmRNA translation when YchF is absent, potentially reduces canonical translation to a level that impairs cell viability at low temperatures.

## 5. Summary and Outlook

The available data for human Ola1 and *E. coli* YchF support the hypothesis that Ola1/YchF regulates the proteostasis network by adjusting canonical and non-canonical translation initiation mechanisms in response to metabolic changes and stress conditions (Figure 4). The stress-induced decrease in YchF in *E. coli* and the stress-induced increase in Ola1 in humans favours in both cases an enhanced non-canonical translation initiation of mRNAs encoding for stress-response proteins and thus promote stress resistance. The differences in their mode of action likely reflects the differences in translation initiation between bacteria and eukaryotes [87,99]. So far, there are no data addressing the function of YchF in archaea, but such studies would be particularly revealing because lmRNAs are rather frequent in archaea and their translation initiation mechanism differs from the mechanisms described in eukaryotes and bacteria [109].

The potential role of YchF/Ola1 in proteolysis is less defined but supported by the observed interaction of Ola1 with subunits of the proteasome and by increased degradation of Hsp70 in the absence of Ola1 [38,41,69]. In bacteria, strong evidence for a role of YchF in proteolysis is still missing and requires further analysis. However, it is also important to note that the cellular concentration of YchF in bacteria is much lower than the Ola1 concentration in eukaryotes [36,129]. Thus, it is possible that the strictly ribosome-associated function of YchF in bacteria evolved into a more general regulatory function of Ola1 in eukaryotes. This would also explain the multiple Ola1-related phenotypes that have been observed in human cells [16,39,57,60,69,74].

The exact function of ATP hydrolysis by YchF/Ola1 and the role of its post-translational modifications, such as phosphorylation of the conserved serine residue (Ser16 in *E. coli*), are also still unknown. ATP hydrolysis is clearly linked to its ribosome binding and to stress response [20,26], but the consequences still need to be analysed in detail. Furthermore, it is unknown why Ola1/YchF have evolved into ATP-hydrolysing enzymes despite having their evolutionary origin within the GTPase family [16,17]. *E. coli* YchF and human Ola1 preferentially hydrolyse ATP over GTP [19,20], although GTP hydrolysis by human Ola1 has also been reported [39]. Hydrolysis of both ATP and GTP has also been shown for the YchF homologue in rice [22,56]. Using preferentially ATP rather than GTP is possibly explained by the increase in the cellular ATP levels during stress conditions, due to the inhibition of metabolic processes [130]. ATP accumulation potentially also increases the levels of the alarmone (p)ppApp [12,131], which acts similar to (p)ppGpp as a stress-signalling nucleotide, but with different regulatory effects [132]. Competitive inhibition of YchF by increasing (p)ppApp levels could help to fine-tune stress-response pathways. Finally, since GTP is highly sensitive to oxidation [133], using the less sensitive ATP might also be important for adjusting translation during oxidative stress conditions.

Despite the high sequence conservation, studies on YchF/Ola1 in different species failed to reveal a comprehensive functional model up to now. Nevertheless, recent data in *E. coli* and humans support a role in the regulation of non-canonical translation initiation, which can serve as a basis for future studies. Considering the documented connection between Ola1 levels and cancer, these studies will be particularly important for developing Ola1 as a potential therapeutic target.

## Figures and Tables

**Figure 1 microorganisms-10-00014-f001:**
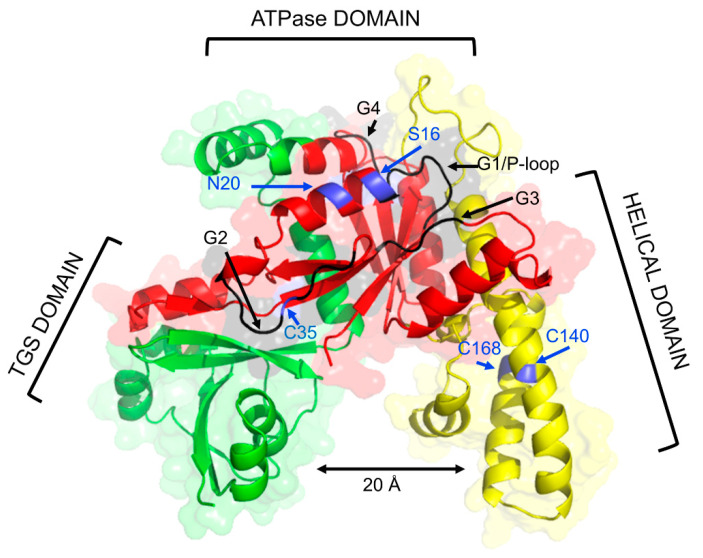
Structural model of *E. coli* YchF. The structure was retrieved from the Alphafold Protein Structure Databank (https://alphafold.ebi.ac.uk/, accessed on 27 November 2021). The ATPase domain is shown in red, the coiled-coil or helical domain in yellow and the TGS-domain in green. Indicated are the four G-motifs (G1–G4), which define and specify nucleotide binding, residue S16, which is phosphorylated, residue N20, which has been mainly used for in vivo site directed crosslinking and residue C35 that is involved in YchF dimerization. The helical domain contains two additional Cys residues in close proximity (C140 & C168), which could act as a redox switch.

**Figure 2 microorganisms-10-00014-f002:**
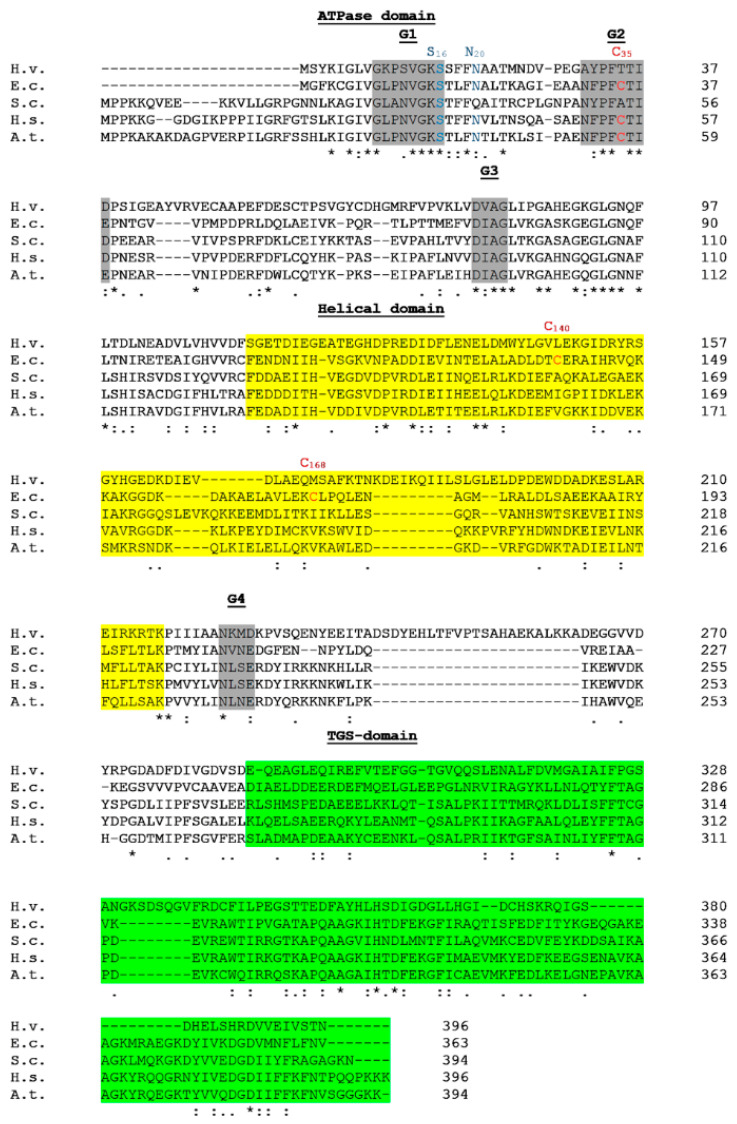
Sequence alignment of YchF/Ola1 from different species. Indicated are the three domains and the G1–G4-motifs, which define nucleotide binding and specificity. In addition, the conserved serine residue (S16 in *E. coli*), which was shown to be phosphorylated and the largely conserved cysteine residue (C35 in *E. coli*), which forms intermolecular disulphide-bridges, are indicated. YchF in *E. coli* and many other proteobacteria contains two additional closely spaced cysteine residues (C140 and C168 in *E. coli*), which could constitute a redox switch. H.v., *Haloferax volcanii*; E.c., *Escherichia coli*; S.c., *Saccharomyces cerevisiae*; H.s., *Homo sapiens*; A.t., *Arabidopsis thaliana*. Alignment was performed with the ClustalW multiple sequence alignment tool. (*) indicates positions which have a single, fully conserved residue; (:) indicates conservation between groups of strongly similar properties (>0.5 in the Gonnet PAM 250 matrix) and (.) indicates conservation between groups of weakly similar properties (≤0.5 in the Gonnet PAM 250 matrix).

**Figure 3 microorganisms-10-00014-f003:**
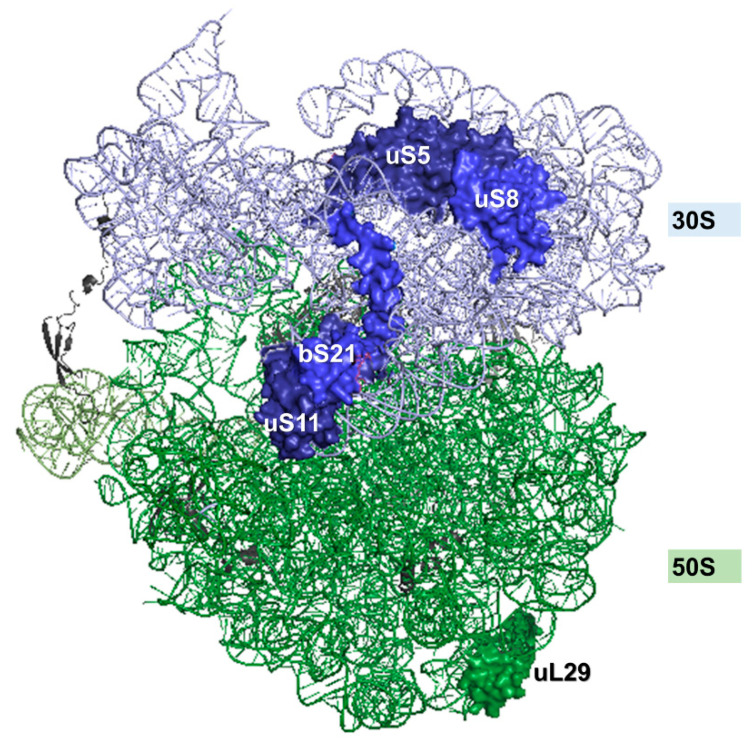
Contact sites for YchF on the *E. coli* 70S ribosome. Contact sites were determined by site-directed in vivo cross-linking combined with mass spectrometry [42]. The 30S ribosomal subunit is shown in light blue and ribosomal proteins that were identified as contact sites in dark blue. The 50S ribosomal subunit is shown in light green and the ribosomal protein uL29, which was also found as contact site is indicated. The 70S ribosome structure was retrieved from the protein data bank with the PDB code: 5mdz [44].

**Figure 4 microorganisms-10-00014-f004:**
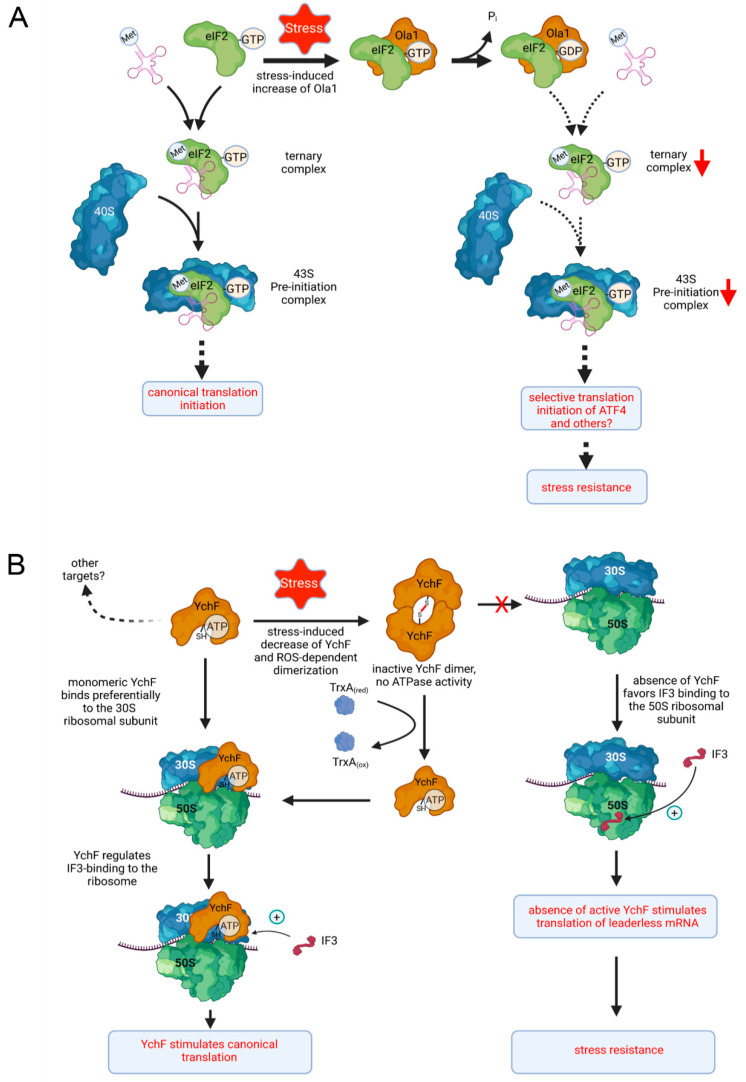
Model of translational regulation by Ola1/YchF. (**A**) In eukaryotes, Ola1 binds to eIF2 and stabilizes eIF2 in its GDP-bound state, which would subsequently reduce the formation of the ternary complex. As a consequence, alternative translation initiation mechanisms are favoured, leading to selective synthesis of stress-response proteins. (**B**) In prokaryotes, such as *E. coli*, YchF binds preferentially to the 30S ribosomal subunit and regulates IF3-binding to the 30S subunit, which favours canonical translation initiation. Upon stress conditions, *ychF* expression is reduced and the ATPase activity of YchF is further inhibited by dimerization. The absence of active YchF possibly favours IF3 binding to the 50S subunit and stimulates leaderless mRNA translation. Note that YchF was also found in contact with the 50S ribosomal subunit; this interaction is likely only transient and not depicted here, although it might be involved in regulating IF3 binding to the ribosome. TrxA corresponds to thioredoxin A, which was shown to dissociate the inactive YchF dimer into active YchF monomers [26]. See text for further details.

## Data Availability

Not applicable.

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
