# Peer review of "The Role of the Universally Conserved ATPase YchF/Ola1 in Translation Regulation during Cellular Stress"

_microorganisms, 2021, doi:10.3390/microorganisms10010014_

Round 1

Reviewer 1 Report

Eukaryotic Ola1 and bacteral YchF are universally conserved stress response proteins acting on the ribosome.  However, a comprehensive functional model for these proteins has not been elucidated.  This review is an extensive overview of recent important research topics on Ola1/YchF.  The authors introduce data concerning on a) the structure of Ola1/YchF and their possible binding sites on the ribosome, b) their cellular roles on stress response, and c) their roles in translation initiation.  From these available data, the authors support the hypothesis that Ola1/YchF regulates the proteostasis network by adjusting canonical and non-canonical translation initiation mechanisms in response to metabolic changes and stress conditions.

This review article is well written, and I have only two minor comments, described below.

1)  Figure 3 depicts simultaneous binding of two molecules of YchF-ATP to the ribosome.  On the other hand, it is described on page 9 (line 327) that "IF3 and YchF occupy overlapping binding sites on the 30S subunit".  Readers may be confused with these expressions.  If you have some information to suggest the simultaneous ribosome binding of two molecules of YchF, you should cite it.  Otherwise, you need to modify some parts of the figure or to add a comment to the legend.

2)  In this review, the authors summarize the functional similarities and differences between bacterial YchF and eukaryotic Ola1.  Amino acid sequence alignment of these proteins from multiple species will help to better understand the content.  I would recommend adding the alignment data, if possible.

Author Response

We thank the reviewer for his positive evaluation. In the revised version, we have addressed the two minor issues:
1. We modified the original Figure 3 (now Fig. 4) and also added a more detailed legend to this figure for explaining better the binding of YchF to the 30S and 50S ribosomal subunits

2. We have included a sequence alignment as new Figure 2.

Reviewer 2 Report

This paper reviews the function of the universally conserved ATPase YchF/Ola1 in cellular stress. It is upon the whole very informative and well-written, so that I believe it will be useful to the investigators working in the field. I have just some suggestions that may help the authors to further improve the manuscript.

Firstly, I got somewhat confused with the seemingly contradictory evidence on the phenotypes deriving from YchF/Ola1 depletion. I understand that this sometimes results in increased, sometimes in decreased, resistance to stress conditions. Perhaps everything would get clearer by discussing separately the phenotypic effects for the two proteins, i.e. first YchF and then Ola1. As it is, the reader has to jump between two different factors so that it becomes difficult to remember which is which and what are the effects of each one.

The discussion about YchF/Ola1 effects on translation initiation is more straightforward. Here, I understand that YchF/Ola1 enhances non-canonical initiation thereby promoting the synthesis of stress-related proteins, using different mechanisms in bacteria and eukaryotes.  However, as regards YchF, I am puzzled by the heavy stress put by the authors on Shine-Dalgarno (SD)-dependent translation versus leaderless translation. Reading the text, one gets the impression that bacterial mRNAs either possess a SD sequence or are leaderless, which is not the case. There is quite a number of prokaryotic mRNAs that do have a leader sequence but no clearly recognizable SD motif (if we agree in calling "leaderless" a mRNA that has the start codon right at the 5' end). Therefore, the statement made by the authors,that "due to the absence of a SD sequence, the AUG codon serves as a major signal...." (line 311) is incorrect unless referred to a limited number of special cases in which cleavage of the (SD-containing) leader sequence does indeed produce a lmRNA. This point should be clarified.

Finally, note that the legend to Fig. 3 is wrong: A) should be "eukaryotes" and B) "prokaryotes"

Author Response

We thank the reviewer for his positive comments. In the revised version, we have addressed the helpful suggestions:

  1. We re-organized the manuscript to make differences in the phenotypes of YchF deletions and Ola deletions/depletions more clear. A version of the revised manuscript in which all changes are highlighted is uploaded as Supplementary File.
  2. We completely agree with the reviewer that some bacterial mRNAs do contain a leader sequence but no clearly detectable SD-motif. We have added this important infromation to the revised mansucript. The reason why we focused on lmRNAs that start with the AUG codon is that sofar only those mRNA species were analyzed in respect to YchF function.
  3. Our mistake in the legend to the original Fig. 3 (now Fig. 4) was corrected

Reviewer 3 Report

The review article by Landwerh et al is focused on the role of the conserved ATPase YchF/Ola1 in translation, particularly under stress conditions. The review begins with the authors briefly describing the available structural model of the E. coli YchF and then moving on to describe the results of various studies suggesting the links between YchF and the ribosome. Next, the authors provide an overview of studies pointing to stress-related functions of YchF/Ola1 and its potential roles during translation. The precise molecular function of YchF/Ola1 is still unknown. Overall, the review provides a useful compendium of information accumulated in the literature regarding these interesting proteins. The article is well written, provides a reasonable level of detail, and is easy to follow.

I have just a couple of minor points:

Line 247. A small misstatement: "a ternary complex, consisting of eIF2, methionine and the tRNAiMet". Ternary here refers to the translation initiation factor eIF2 forming a complex with GTP and the aminoacylated Met-tRNAi (that is, eIF2 – GTP – Met-tRNAiMet).

Line 295. To avoid misinterpretation of this sentence, it would be better to say, "those mRNAs that lack the Shine-Dalgarno sequence".

Line 365. Why is OLA1 in all caps here?

Author Response

We thank the reviewer for the positive comments on our manuscript. In the revised version, we have addressed the three minor issues:

  1. We corrected our mistake about the nature of the ternary complex.
  2. We have added `those´to the sentence.
  3. We have corrected OLA1 to Ola1.